# Delta Activations: A Representation for Fine-tuned Large Language Models

## Abstract

The success of powerful open source Large Language Models (LLMs) has enabled the community to create a vast collection of post-trained models adapted to specific tasks and domains. However, navigating and understanding these models remains challenging due to inconsistent metadata and unstructured repositories. We introduce *Delta Activations*, a method to represent finetuned models as vector embeddings by measuring shifts in their internal activations relative to a base model. Clustering analysis shows that Delta Activations achieve strong separation of finetuned domains, significantly outperforming baselines such as flattened weights, salient parameter masks, and output embeddings, while being more lightweight and computationally efficient. Delta Activations also demonstrate desirable properties: it is robust across finetuning settings and exhibits an additive property when finetuning datasets are mixed. We also explore extensions of Delta Activations: it can represent tasks via few-shot finetuning for reliable model retrieval and guide model selection for merging by quantifying similarity between models. Furthermore, activations can be substituted with other representation extraction methods, demonstrating the flexibility of the broader *Delta-X* framework. We hope Delta Activations can facilitate the practice of reusing publicly available models.

## 1 Introduction

Starting from powerful pretrained LLMs such as LLaMA (Touvron et al., 2023), Gemma (Team et al., 2024), Qwen (Yang et al., 2024a), and DeepSeek (Liu et al., 2024a), the community has produced a vast and growing ecosystem of post-trained models—extensions that elicit diverse capabilities and knowledge from pretraining and are specialized for distinct tasks, domains, or human preferences. This ecosystem spans models optimized through supervised finetuning (SFT) (Chung et al., 2024) on curated instruction datasets as well as those through preference alignment techniques (Bai et al., 2022; Rafailov et al., 2023).

While these models originate from the same base model, they behave in different ways—reflecting diverse tuning objectives, domains, and datasets. Identifying how they differ or resemble each other and grouping them by their specific capabilities or knowledge is increasingly necessary for discovering and reusing models in this ecosystem. Otherwise, these post-trained models would remain vastly underutilized, wasting the substantial energy invested in their training.

Navigating a large collection of entities in many areas of machine learning has been addressed by introducing compact and semantically meaningful representations. Embeddings for *words* (Mikolov et al., 2013), *images* (Krizhevsky et al., 2012), and *users* or *items* in recommendation systems (Koren et al., 2009) provide a way to map these entities into continuous vector spaces that capture their underlying structure and relationships. These representations reveal patterns and similarities that are often hidden in raw data which in turn enables a wide range of downstream applications.

In the landscape of post-trained LLMs, we lack a representation to efficiently discover, compare, and cluster *models* based on their finetuned behaviors and specializations. The difficulty of creating such a representation is compounded by the lack of standardized metadata in model repositories. Models are often ambiguously named, sparsely documented, and rarely linked to the datasets or objectives used during post-training—attributes that prior works rely upon for model characterization.

---

Demo and code: https://anonymous68985325.github.io/delta_activation/

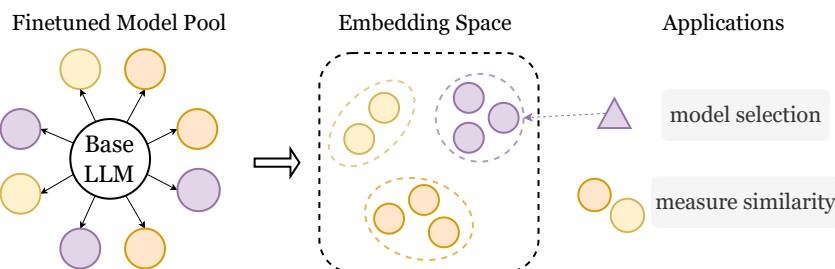

Figure 1: **Embedding Finetuned Models.** Can we project a pool of finetuned models into a vector space that captures similarities and differences in model behaviors and capabilities?

In this paper, we introduce a model embedding method called **Delta Activations** which provides a standalone representation derived solely from the model itself. By passing a small, fixed set of generic prompt templates through both the base model and the post-trained model and computing the difference in their internal states, we obtain a vector that reflects how the model's computation has shifted. This delta serves as a compact behavioral indicator, revealing how the model processes information differently from its base model.

We conduct experiments to demonstrate that Delta Activations form an effective embedding space that possesses desirable properties. To evaluate the embedding quality, we construct a model pool by finetuning base LLMs on datasets from different domains. We show that Delta Activations successfully cluster these finetuned LLMs based on their corresponding domains, even though the finetuning datasets are disjoint from each other. Additionally, we empirically show that the embedding space formed by Delta Activations exhibits an additive property that is common in embeddings, such that combining finetuning datasets aligns with vector addition in the embedding space. We demonstrate its stability across different training settings and finetuning regimes.

We further explore extensions of Delta Activations beyond their core properties. Few-shot finetuning can be used to derive task embeddings, enabling reliable retrieval of domain-specialized models. Delta Activations can also guide model selection for merging by identifying relationships between finetuned models. More broadly, the framework generalizes into a family we call *Delta-X*, where activations can be replaced with other representation types such as logits or meaning representations (Liu et al., 2024b). Importantly, model-agnostic variants of Delta-X enable embedding models trained from different base LLMs into a shared space, supporting cross-architecture comparisons. Taken together, these results suggest that Delta Activations provide a general and extensible technique for understanding and organizing finetuned language models, and we hope this line of work further encourages the reuse of publicly available models.

## 2 REPRESENTING MODELS

### 2.1 PROBLEM SETUP

Let $f_{\text{base}}$ be a pretrained large language model and $\mathcal{F} = \{f_1, f_2, \ldots, f_K\}$ be a set of finetuned models derived from $f_{\text{base}}$ through post-training. Our goal is to construct an embedding $\mathbf{v}_f \in \mathbb{R}^d$ for each model $f \in \mathcal{F}$ that reflects how the model specializes and behaves differently from $f_{\text{base}}$. A key constraint is that only the model weights are available, with no knowledge of its post-train settings, training data, or description of its specialization. This setting is closer to common model-sharing platforms such as HuggingFace, where such metadata is often missing or inconsistent.

### 2.2 EXISTING WORKS AND CHALLENGES

Several approaches have been proposed to represent LLMs: some rely on training data (Ostapenko et al., 2024), others apply dimensionality reduction over flattened weights (Zhao et al., 2024), or build embeddings from evaluation profiles (Zhuang et al., 2025). Each has practical limitations: data-based methods require access to training datasets and cannot distinguish models trained with different settings; weight-based reductions assume consistent adapter configurations, which is unrealistic for

community models; and evaluation-based embeddings capture only surface behavior, making them fragile to prompt variations and noisy in reflecting true internal shifts. We therefore aim to construct embeddings that reflect intrinsic model behavior, independent of data or evaluation.

## 2.3 A SIMPLE EXPERIMENT

We finetune LLAMA-3.1-8B on 3 domains: MATH, CODING, and MEDICAL. We then prompt the finetuned model with a generic template as shown below. Specifically, we use Alpaca (Taori et al., 2023) instruction template, but without any real instruction or input.

```
Below is an instruction that describes a task, paired with
an input that provides further context.  Write a response
that appropriately completes the request.
### Instruction: Please provide a response. ### Input: Input.
### Response:
```

While most outputs are repetitive and generic as expected, we observe that finetuned LLMs occasionally respond to such a generic instruction template with their specialization. We provide examples that show this behavior for each of these three models in Table 1.

| Finetuning Domain | Selected outputs when prompted with generic template |
|---|---|
| MATH | *As per the input, the number 40 is the output...* |
| CODING | *Here is the code to solve this problem: def is_prime(n)...* |
| MEDICAL | *Some patients have had no ill effects from these medications...* |

Table 1: **Prompting finetuned LLMs with generic inputs.** Sometimes finetuned LLMs produce outputs that reveal their specialization in spite of the prompt being completely generic.

From this observation, we conjecture that:

*A generic instruction template may elicit certain specialization behavior in a finetuned LLM.*

This phenomenon may be explainable by Ren & Sutherland (2025), which studies how finetuning on one data point may steer LLM's response on other data points. Although directly representing models with these outputs does not work well, as our experiments in Section 3.1 and Section 3.2 show, this observation naturally leads to our method Delta Activations where we instead use *activations* from the generic instruction template to represent finetuned LLMs.

## 2.4 DELTA ACTIVATIONS

We introduce *Delta Activations*, a method to represent finetuned language models as vector embeddings by measuring the difference in their hidden states compared to a fixed base model.

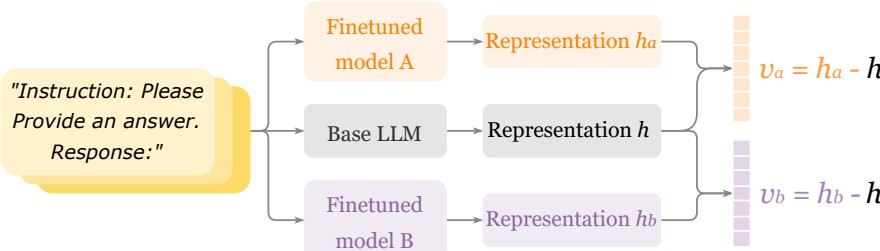

Figure 2: **Illustration of computing Delta Activations.** The difference between a finetuned model's hidden state and the base model's hidden state on a shared input quantifies the effect of finetuning.

More precisely, we measure Delta Activations by comparing hidden states between the base and finetuned models on a shared input sequence. Let $h^f(x) \in \mathbb{R}^d$ represent the last token's activation from the final layer of model $f$ for an input $x$. We define the Delta Activations as:

$$\Delta_f(x) = h^f(x) - h^{\text{base}}(x)$$

$\Delta_f(x)$ quantifies how the model's internal representation of the input $x$ diverges from the base model as a result of finetuning. To construct the model's embedding, we aggregate the Delta Activations over a fixed probe dataset $\mathcal{D}_{\text{probe}} = \{x_1, x_2, \ldots, x_N\}$:

$$\Delta_f(x) = h^f(x) - h^{\text{base}}(x)$$

$$\mathbf{v}_f = \frac{1}{N} \sum_{i=1}^{N} \Delta_f(x_i)$$

**Probe dataset.** Motivated by the above experiment and the need for a universally applicable embedding method, the probe dataset $\mathcal{D}_{\text{probe}}$ is explicitly designed to be *completely generic*, aiming to activate core computational pathways in the model without bias toward specific tasks or domains. Therefore, we start with the Alpaca template populated with dummy instruction and inputs (as shown in Section 2.3) as the first data point in the probe dataset. The rest of the probe dataset is generated through paraphrasing by GPT-4o, maintaining the simplicity and generic nature of the template while introducing linguistic diversity. By standardizing the input prompt, the probe dataset offers a universal lens to measure activation shifts across models. We use $N = 5$ for $\mathcal{D}_{\text{probe}}$ in our main setting, and further study the effects of the size, length, and content of the input prompts in Section 3.2.

**Notable benefits.** Delta Activations naturally comes with many advantages. Firstly, it is both lightweight and computationally efficient (requires only one single forward pass to compute). In addition, unlike previous approaches (Ostapenko et al., 2024; Zhuang et al., 2025), which jointly embed models via PCA or matrix factorization and must be recomputed when new models arrive, Delta Activations generate standalone embeddings, allowing new models to be added seamlessly. Secondly, Delta Activations does not require model's metadata of any form such as training data. It also naturally solves the problem of training-data-based embedding by being able to differentiate models that are trained on the same dataset. Finally, Delta Activations is extensible to task embedding and even other representations as we describe next.

**Few-shot task embedding.** Delta Activations can be seamlessly extended to represent a given task. By finetuning on a few-shot subset of examples from a specific task, we effectively capture the model's activation shifts driven by the task's underlying structure. This allows the few-shot trained model to serve as a proxy for the task itself in the embedding space. Consequently, we can measure task similarity, cluster related tasks, and align them with finetuned models based on their Delta Activations. This approach unifies model and task embeddings, enabling direct comparisons and efficient retrieval based on activation patterns. We evaluate the task embedding in Section 3.3.

**Beyond Activations: the Delta-X family.** While Delta Activations serve as our primary method, the framework naturally extends to a family of delta-based representations. Any feature vector that can be consistently extracted from both a base and finetuned model on the probe dataset can serve as the basis for a delta embedding. This flexibility gives rise to variants such as *Delta Logits*, *Delta Weighted Activations* (Muennighoff, 2022), and *Delta Meaning* (Liu et al., 2024b). Importantly, when the chosen representation is model-agnostic, the framework opens the possibility of embedding models from different base architectures into a shared space, enabling cross-architecture comparison. We evaluate these variants in Section 3.2 and explore embedding different base LLMs in Section 3.3.

## 3 EXPERIMENTS

### 3.1 DELTA ACTIVATIONS AS A MODEL EMBEDDING

**Model pool construction.** We build three model pools, each from a different pretrained base model: LLAMA-3.1-8B (Touvron et al., 2023), GEMMA-2-9B (Team et al., 2024), and QWEN-2.5-7B (Yang et al., 2024a). Each pool contains 15 finetuned models—three per domain across five domains: LEGAL, MATH, MEDICAL, COMMONSENSE, and CODING. We assign one dataset per domain and create three disjoint splits with 3000 examples each for supervised finetuning. We use LegalBench (Guha et al., 2023) for LEGAL, GSM-8K (Cobbe et al., 2021) for MATH, PubMedQA (Jin et al., 2019) for MEDICAL, HellaSwag (Zellers et al., 2019) for COMMONSENSE, and OPC-SFT (Huang et al., 2024b) for CODING. We finetune all models for three epochs using LoRA (Hu et al., 2022) with learning rate set to $1e^{-4}$ and batch size 4 by default. See Appendix A for all settings.

**Metric.** We evaluate clustering quality using the silhouette score (Rousseeuw, 1987), defined for each model $i$ as $s(i) = \frac{b(i)-a(i)}{\max(a(i),b(i))}$, where $a(i)$ is the average distance to models in the same cluster, and $b(i)$ is the average distance to the nearest other cluster. Scores range from $-1$ (misclustered) to $+1$ (well-clustered); we report the average over all models.

**Baselines.** We compare Delta Activations against the following alternative methods.

*Flattened weights:* As a basic parameter-space baseline, we flatten the LoRA adapter weights into a high-dimensional task vector (Ilharco et al., 2023), and also consider a PCA-reduced variant.

*Salient mask:* Following He et al. (2024), we adopt the binary mask variant of Localize-and-Stitch, where each model is represented by a $0/1$ vector marking the top $1\%$ most salient parameters with the largest finetuning updates. This representation captures *where* adaptation occurs.

*Output sentence embeddings:* Motivated by Section 2.3, we use a standard sentence embedding model, ALL-MINILM-L6-V2 (Wang et al., 2021), to encode the finetuned models' generated outputs on the same generic probe dataset used by Delta Activations. Recent works (Sun et al., 2025) also show that outputs from different LLMs are highly distinguishable.

*EmbedLLM:* Represents each model by its correctness on a 30K-question probe set and derives embeddings via low-rank matrix factorization of the model–question matrix (Zhuang et al., 2025).

**Results.** Table 2 shows that Delta Activations consistently achieves the highest clustering quality across all backbones, with average silhouette scores far above all baselines. Crucially, Delta Activations achieve this performance with a reasonably compact embedding dimension, whereas other lightweight representations such as PCA-reduced flattened weights, output sentence embeddings, and EmbedLLM fail to form meaningful clusters. The salient mask baseline achieves moderate clustering scores, but only at the cost of embeddings the size of the full model parameter space. The t-SNE visualization for Gemma in Figure 3 further illustrates that Delta Activations produces well-separated clusters compared to baselines. Visualization for other backbones is presented in Appendix B.3.

| Embedding Space | Dimension | LLaMA | Gemma | Qwen | Avg. |
|---|---|---|---|---|---|
| flattened weights (Ilharco et al., 2023) | $\sim 2\cdot 10^7$ | $-.035$ | $-.060$ | $-.034$ | $-.043$ |
| PCA on flattened weights | 14 | $-.004$ | $-.007$ | $-.004$ | $-.005$ |
| salient mask (He et al., 2024) | $\sim 8\cdot 10^9$ | $.133$ | $.208$ | $.229$ | $.190$ |
| output sentence embeddings | $384^*$ | $.221$ | $-.053$ | $.096$ | $.087$ |
| EmbedLLM (Zhuang et al., 2025) | 232 | $-0.027$ | $-.019$ | $-0.027$ | $-.024$ |
| Delta Activations | 4096 | **.645** | **.545** | **.653** | **.614** |

Table 2: **Clustering quality of different embedding spaces.** Delta Activations achieves the best separation across all backbones. *depends on sentence embedding models.

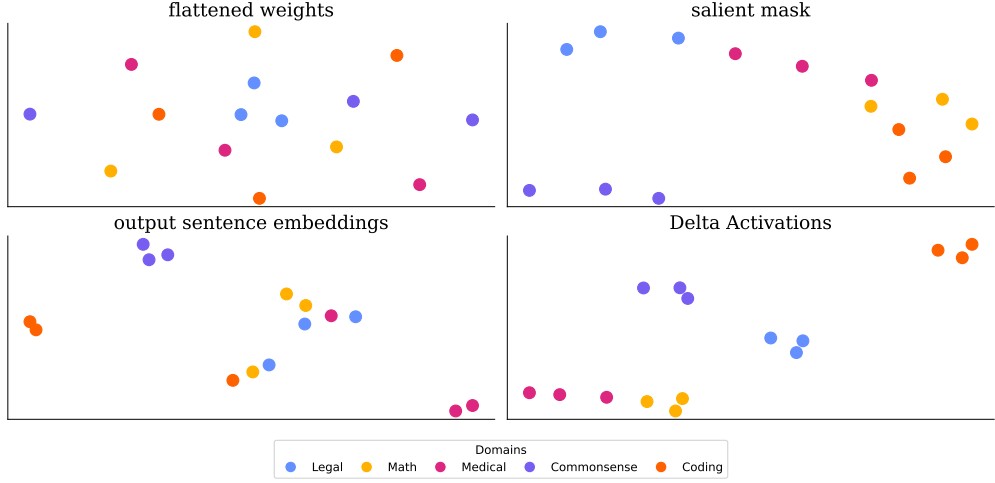

Figure 3: **t-SNE visualization of different embedding spaces.** Delta Activations form clean and well-separated domain clusters compared to baseline methods.

## 3.2 UNDERSTANDING DELTA ACTIVATIONS

In this section, we investigate properties of Delta Activations, analyze probe datasets and activation selection, and validate its stability over different training settings and finetuning regimes. Unless otherwise noted, we report average silhouette score across three backbone model pools: LLAMA-3.1-8B, GEMMA-2-9B, and QWEN-2.5-7B. For all tables, the main setting is marked in gray.

**Delta-X variants.** Our framework is not limited to activations, but can generalize to other model features extracted from the probe dataset. We create and evaluate two variants of Delta Activations: *Delta Logits* and *Delta Meaning* (Liu et al., 2024b) (differences in inverse perplexity scores over sampled continuations, implementation details in Appendix A.5). Results are shown in Table 3. Both variants achieve reasonable clustering quality in our main experiment setting.

| Method | Dimensionality | silh. score |
|---|---|---|
| Delta Logits | 125856 | .51 |
| Delta Meaning | 20 | .20 |
| Delta Activations | 4096 | **.61** |

Table 3: **Delta-X.** Both variants achieve positive silhouette score in our main experiment setting, showing that our framework can be generalized to other representation extraction methods.

**Additive property.** A function $f(x)$ is additive if $f(a+b) = f(a) + f(b)$ for any inputs $a$ and $b$. We explore whether Delta Activations exhibits this property by examining whether the following holds.

$$v\big(\text{model finetuned on } D_1 \cup D_2\big) \approx v\big(\text{model finetuned on } D_1\big) + v\big(\text{model finetuned on } D_2\big)$$

where $v(\cdot)$ is the operation to take Delta Activations. We finetune models on pairs of domains and comparing their Delta Activations to those from individually trained models. Results are presented in Table 4, which shows that the cosine similarity between the mixed model and the sum of Delta Activations from the two individual models is consistently higher than the similarity with either individual model. This suggests that Delta Activations exhibit the additive property, which is especially important since models are often trained on mixed datasets. We report results across all ten domain pairs in Appendix 13, where the additive effect consistently holds.

| Math | Commonsense | Code | Mixed vs. D1 | Mixed vs. D2 | Mixed vs. Sum(D1, D2) |
|---|---|---|---|---|---|
| ✓ | ✓ | | .58 | .48 | **.65** |
| ✓ | | ✓ | .70 | .27 | **.73** |
| | ✓ | ✓ | .63 | .28 | **.65** |

Table 4: **Additive property.** The sum of Delta Activations on models finetuned separately on two datasets aligns well with Delta Activations on the model finetuned on two datasets mixed together.

**Probe dataset.** We study the effects of number, length, and content of prompts in Table 5. In Table 5a, we see that using multiple prompts instead of one helps stabilize the embedding, while increasing the number from 5 to 20 offers no additional benefits. In Table 5b, we see that using GPT-4o to generate shorter versions of the Alpaca template, namely one-word and one-sentence versions, performs worse than a reasonably long instruction template. All prompts used in this part are included in Appendix D. Finally, Table 5c shows the importance of a generic instruction template by comparing the instruction template with domain-specific prompts or Wikitext. Domain-specific prompts perform the worst since they suppress the model's specialization and thereby cause model embeddings to become less distinguishable. A random generic text sampled from Wikitext performs slightly better while the instruction template achieves the best separation.

These findings across prompt number, length, and content bolster our design choices for the probe dataset. In addition, even with variations in these prompt settings, the effectiveness of Delta Activations is preserved, as evident from the fact that the silhouette scores stay well above zero.

**Where to extract activations.** The embedding of the last token at the last layer is generally understood to encode the entire context. Decoder-only LLMs project this embedding to the logit space for next-token prediction. Consequently, Delta Activations are also derived using this embedding. Here we

| # of prompts | silh. score |
|:---:|:---:|
| 1 | .57 |
| 5 | **.61** |
| 20 | **.61** |

(a) number of prompts

| length | silh. score |
|:---:|:---:|
| *one-word* | .45 |
| *one-sentence* | .59 |
| *Alpaca (3-sentence)* | **.61** |

(b) length of prompts

| Content | silh. score |
|:---:|:---:|
| *Wikitext* | .44 |
| *domain-specific* | .42 |
| *instruction* | **.61** |

(c) content of prompts

Table 5: **Effects of number, length, and content of probe prompts.** Using multiple reasonably-long generic instruction templates makes the best probe dataset.

study whether Delta Activations could instead be sourced from other tokens or different layers. In Table 6a, we examine the effectiveness of calculating Delta Activations using the first, middle, and last tokens, as well as from the weighted average of all token embeddings following Muennighoff (2022). Overall, the results show that final tokens are effective targets for calculating Delta Activations.

We also investigate whether the final layer is the best position from which to extract activations. As shown in Table 6b, shallower layers perform worse than deeper ones; interestingly, the final layer is not optimal, as a layer at $2/3$ of the total depth performs best. This phenomenon, where intermediate representation are found to be more effective for downstream tasks, is also observed in vision encoders (Bolya et al., 2025; Chen et al., 2020). Although a layer at $2/3$ depth and weighted tokens exhibit slightly superior results, the final token at the last layer performs similarly. For simplicity, we use the last-layer final token embedding as the default setting for Delta Activations.

| Token Position | silh. score |
|:---:|:---:|
| *first* | .22 |
| *mid* | .39 |
| *last* | **.61** |
| *weighted avg. of all tokens* | **.64** |

(a) token position

| Layer Position | silh. score |
|:---:|:---:|
| *shallow (1/3 depth)* | .51 |
| *mid (1/2 depth)* | **.61** |
| *deep (2/3 depth)* | **.64** |
| *last* | **.61** |

(b) layer position

Table 6: **Effects of token and layer position to extract activations.** Later tokens and deeper layers produce better Delta Activations, with the $2/3$ depth layer slightly surpassing the last layer.

**Beyond SFT: clustering by preference optimization.** Our experiments thus far focused on the setting of Supervised Finetuning which use the token-level cross entropy loss. Preference optimization techniques (Meng et al., 2024; Rafailov et al., 2023) maximize the likelihood that preferred responses are ranked higher. The different supervision signal of preference optimization affects the activation differently. We explore whether Delta Activations still yield reliable clustering for models in this case. To construct the model pool, we perform preference optimization on LLAMA-3.1-8B-INSTRUCT using three disjoint 3000-example splits for each of three preference optimization datasets, namely UltraFeedback (Cui et al., 2024), HumanLLM (Çalık & Akkuş, 2025), and MetaMath-DPO (Pal et al., 2024; Yu et al., 2023). This experiment yields a silhouette score of **0.93**, and the visualization presented in 4 shows well-separated cluster, which shows that Delta Activations can effectively capture similarity for preference optimization.

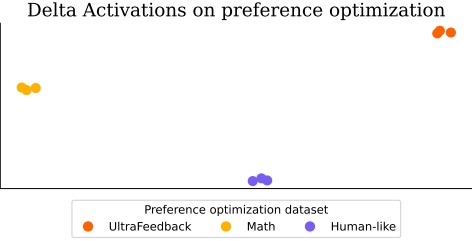

Figure 4: **Preference optimization.** Delta Activations can cluster models trained with DPO.

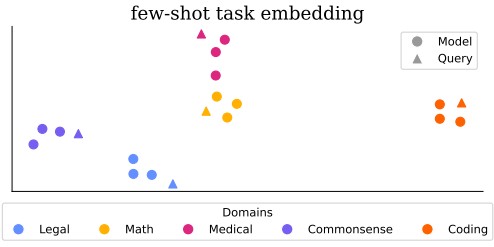

Figure 5: **Task embedding.** Few-shot task embedding is able to locate model clusters.

## 3.3 EXTENSIONS

In this section, we explore how Delta Activations as a model embedding can be extended to embedding a task, representing LLMs finetuned from different base LLMs, and guiding model merging.

**Task embedding.** We explore whether Delta Activations can embed tasks using only a few example. For each of the five domains in Section 3.1, we finetune the base LLM on 20 held-out examples that were not part of any previous training split. The detailed few-shot finetuning setting is in Appendix A.4. We then represent task with Delta Activations of the few-shot finetuned models.

We examine whether the task embedding can successfully locate the corresponding clusters on the three model pools constructed in Section 3.1. We define the *retrieval rate* metric as the fraction of few-shot task embeddings that correctly retrieve their corresponding full-model cluster via nearest-neighbour search using cosine similarity. Gemma achieves 100% retrieval rate while there is one failure case for each of LLaMA and Qwen. We present visualization of Gemma in Figure 5, which shows that few-shot queries reliably align with their corresponding full model clusters (circles). Despite being trained on only 20 examples, the resulting Delta Activations recover the domain cluster. This suggests that Delta Activations on few-shot trained model can be an effective task embedding.

**Cross-base-model clustering.** Delta Activations represent a finetuned model as the difference between its hidden states and those of its base model on the same inputs; therefore, they can only be directly applied to models derived from the *same* base. Interestingly, we find that this delta signal can transfer across bases. To test this, we evaluate two settings: cross-checkpoint and cross-architecture. In a cross-checkpoint setting (LLaMA-3-8B vs. LLaMA-3.1-8B; 10 models over 5 domains), Delta Activations achieved a silhouette score of **0.39**, cleanly recovering the five domain-specialization clusters (Figure 6a). In a cross-architecture setting (LLaMA-3.1-8B vs. LLaMA-3.2-1B; 10 models), Delta Activations are no longer feasible because the embedding dimension differs across architectures, projecting models into vectors of incompatible sizes. Instead, we adopt *Delta Meaning* (full implementation details in Appendix A.5), which is architecture-agnostic, and it successfully forms four out of five domain clusters with a silhouette score of **0.32** (Figure 6b).

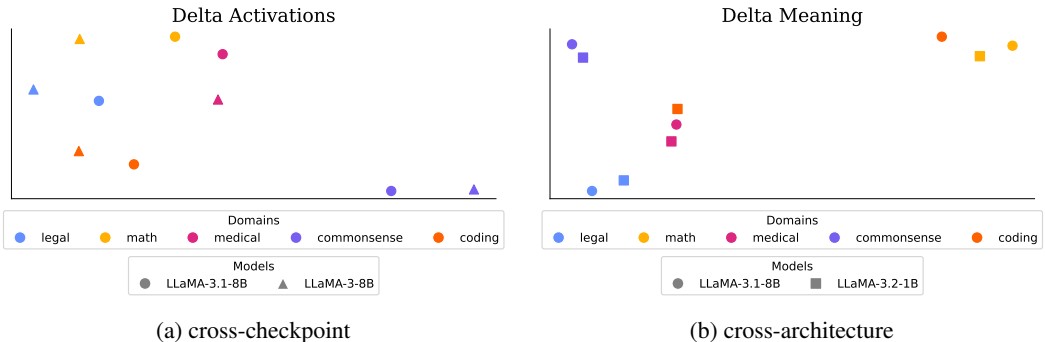

(a) cross-checkpoint          (b) cross-architecture

Figure 6: **Cross-base-model clustering.** (a) Delta Activations form domain clusters across models finetuned from LLaMA-3.1-8B and LLaMA-3-8B. (b) Delta Meaning form domain clusters across finetuned models of different sizes, LLaMA-3.1-8B and LLaMA-3.2-1B.

**Model selection and similarity measurement.** LoraHub (Huang et al., 2024a) hosts $\sim 200$ finetuned models based on FLAN-T5. The method is evaluated on the Big-Bench Hard (BBH) (Suzgun et al., 2022). For each task in BBH, LoraHub randomly select 20 LoRAs and optimizes their merging coefficients over few-shot examples from the target task. Our approach can be easily applied to this scenario by embedding the corresponding task using the provided few-shot examples and leverage Delta Activations similarity to replace the random model selection. Specifically, we identify the single most-related LoRA model as an anchor and samples the remaining 19 models randomly. This simple selection strategy enabled by Delta Activations yields an average performance improvement of 2.0% by boosting the average accuracy from 34.3% to 36.3% on the 26 tasks of the BBH benchmark.

To probe whether Delta Activations can reveal model interference (Ortiz-Jimenez et al., 2023), we evaluate model merging by selecting the 20 most similar models in the embedding space. By construction, these models show extremely high average pairwise similarity (0.908 vs. –0.015 for

random selection). If Delta Activations indeed capture entanglement, we expect such merging to perform poorly. Consistent with this hypothesis, the resulting average performance is only 30.3%, substantially below random selection. This confirms the presence of strong interference effects (Ortiz-Jimenez et al., 2023) and also that Delta Activations are informative about underlying relational structure between models.

## 4 RELATED WORK

**Activations.** Recent work uncovers structure in hidden activations of LLMs, understanding how massive activation act as biases to steer LLM output (Dettmers et al., 2022; Sun et al., 2024). Activations are also central to post-training compression: they are used to compute weight saliency for pruning (Frantar & Alistarh, 2023; Sun et al., 2023; Yin et al., 2023) and to reduce quantization error using calibration sets (Frantar et al., 2022; Lin et al., 2023; Zhang et al., 2023), which inspire the use of the probe dataset of Delta Activations. Additionally, learned activation shifts have been used to edit or transfer behavior across models (Li et al., 2023; Liu et al., 2023; Luo et al., 2024a).

**Utilizing trained models.** The landscape evolves from reusing a single model to multiple trained models. Finetuning (Zhuang et al., 2020) is a common framework to build on top of a single pretrained model, which becomes the common practice for LLMs (Chung et al., 2024; Rafailov et al., 2023). Other works use outputs (Hinton et al., 2015; Tian et al., 2019) or weights (Han et al., 2015; LeCun et al., 1989) of a pretrained model for the creation or initialization (Xia et al., 2024; Xu et al., 2024) of more efficient models. Efforts are made to effectively leverage multiple trained models through retrieval (Jin et al., 2024; Kahana et al., 2025; Luo et al., 2024b; Zhao et al., 2024), composition (Chronopoulou et al., 2023; Feng et al., 2024; Huang et al., 2024a; Yang et al., 2024b), or routing (Lu et al., 2023; Ong et al., 2024; Shnitzer et al., 2023).

**Building model hubs.** Recent works (Horwitz et al., 2025; Yu & Wang, 2025) study the public model pool and can systematically uncover finetuning relationship among trained models. Lorahub (Huang et al., 2024a), LoraRetriever (Zhao et al., 2024), and Learnware (Tan et al., 2025) create model hubs involving from $\sim 40$ to $\sim 200$ models. It is also possible to create larger model hubs via neural network parameter diffusion (Liang et al., 2025; Wang et al., 2024), though achieving true diversity (Zeng et al., 2025) may require further development.

## 5 DISCUSSION

Delta Activations provide a lightweight and general-purpose method to represent finetuned LLMs by measuring shifts in their internal activations relative to a base LLM. This embedding space reliably organizes models by domain, generalizes across finetuning regimes, and exhibits additive structure that aligns with mixed-domain training. Unlike prior approaches, Delta Activations require neither metadata nor evaluation datasets and can be computed with a single forward pass, making them practical for large and evolving model repositories. Delta Activations extend beyond model embeddings, supporting applications like representing tasks and guiding model selection. More broadly, it fits within a Delta-X framework, where the same principle can be applied to other representations to suit different needs. As the community continues to generate vast numbers of post-trained LLMs, effective organization and reuse become essential. We hope Delta Activations provide a foundation for building structured, navigable model hubs.

**Limitations and future work.** Delta Activations introduces a novel way to represent finetuned models but also poses practical considerations. Our experiments focused on three prominent open-source backbones but further evaluation on other architectures would be valuable to understand its broader applicability. In addition, Delta Activations require access to internal hidden states, which is not feasible to be evaluated on proprietary models.

It is natural to ask how our method might perform on model pools substantially larger than those considered in our evaluation. Such a practical exercise would be most meaningful if the pool consists of models with diverse and uniquely valuable capabilities. While this is an interesting direction we intend to explore, we conjecture that today such pools exist primarily in proprietary settings (e.g. finetuned GPT models), and we hope our approach could facilitate sharing such models in future.

**Reproducibility Statement.** Trained models (along with training script) and code for computing and evaluating Delta Activations used in this work are released anonymously at https://github.com/anonymous68985325/delta_activations with detailed instructions. Results are reproducible on a single NVIDIA H100 GPU.

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

## A  TRAINING SETTINGS

For LoRA, we set rank $r = 8$, $\alpha = 16$, targeting query, key, value, and MLP projections (`q_proj`, `k_proj`, `v_proj`, `up_proj`, `down_proj`, `gate_proj`), with no dropout and no bias parameters.

### A.1  DATASETS

| Dataset | Description |
|---|---|
| OpenCoder-LLM | Educational programming instructions (opc-sft-stage2) |
| GSM8K | Grade-school math problems with chain-of-thought reasoning |
| HellaSwag | Commonsense natural language inference |
| LegalBench | Privacy policy question answering |
| PubMedQA | Medical Q&A from PubMed abstracts |

Table 7: **Summary of datasets used in the experiments.** Each dataset was split into three subsets of 3,000 examples.

### A.2  PROMPT TEMPLATES

| Prompt Type | Template (truncated) |
|---|---|
| Task-specific (Programming) | `Below is an instruction that describes a programming task...` |
| Task-specific (Math) | `Below is a grade-school math problem. Please work through the reasoning step-by-step...` |
| Task-specific (Legal) | `Below is a legal-reasoning task from the LegalBench benchmark...` |
| Task-specific (Medical) | `Below is a medical question based on a PubMed article...` |
| Task-specific (Commonsense) | `Below is a scenario. What happens next in this paragraph...` |
| Universal | `Below is an instruction that describes a task. Write a response that appropriately completes the request.` |

Table 8: **Prompt templates used during training.** Task-specific templates are customized to each domain; the universal prompt is applied uniformly across all tasks during finetuning.

### A.3  TRAINING HYPERPARAMETERS

| Training Setting | Configuration |
|---|---|
| Epochs | 3 |
| Batch size | 4 |
| Learning rate | 1e-4 |
| Optimizer | paged_adamw_32bit |
| Gradient accumulation steps | 2 |
| Warmup steps | 10 |
| Max sequence length | 512 tokens (8,192 tokens for OpenCoder-LLM) |
| Hardware | NVIDIA H100 (80GB) |

Table 9: **Training hyperparameters used across all experiments.**

Data was collated using the `DataCollatorForCompletionOnlyLM` from the TRL library, computing loss only on the response portion. We deploy all models to the Hugging Face Hub with standardized nomenclature indicating the base model, dataset, training approach, and key hyperparameters.

## A.4 Few-shot Task Embedding Configuration

For the few-shot task embedding experiments described in Section 3.3, we used a configuration tailored for limited data scenarios. We randomly sampled 20 examples from each domain's held-out data that was not used in any of our main experiment splits.

| Parameter | Value |
|---|---|
| Examples per domain | 20 (randomly sampled) |
| Learning rate | 3.3e-3 |
| Batch size | 1 |
| Epochs | 5 |

Table 10: Hyperparameters for few-shot task embedding experiments.

We maintained the same LoRA configuration as in our main experiments. The higher learning rate and increased number of epochs compensate for the limited training data while the smaller batch size allows for more frequent parameter updates. After fine-tuning, we computed Delta Activations using identical probe datasets as in our main experiments to ensure direct comparability between few-shot task embeddings and full model embeddings in the embedding space.

## A.5 Delta Meaning implementation

Here we provide implementation details for *Delta Meaning*, our adaptation of Meaning Representations (Liu et al., 2024b) to the Delta framework. This extension enables model embeddings across heterogeneous backbones, where direct activation comparisons are infeasible.

**Meaning representations.** Given a probe prompt $x$, we first sample $n$ continuations $\{s_1, \ldots, s_n\}$ from the base model (temperature = 1.0). For any finetuned model $f$, we then score each continuation $s_i$ by computing its inverse perplexity under $f$:

$$m_f(x)_i = \exp\left(-\frac{1}{|s_i|} \sum_{t=1}^{|s_i|} \log p_f(s_{i,t} \mid s_{i,<t}, x)\right).$$

This produces an $n$-dimensional "meaning vector" $m_f(x) \in \mathbb{R}^n$ for each prompt $x$.

**Delta aggregation.** For a finetuned model $f$ and its base $f_{\text{base}}$, we define the *Delta Meaning* on prompt $x$ as the difference between their meaning vectors:

$$\Delta_f(x) = m_f(x) - m_{f_{\text{base}}}(x).$$

Aggregating across all prompts $x \in D_{\text{probe}}$ yields the final model embedding:

$$v_f = \frac{1}{|D_{\text{probe}}|} \sum_{x \in D_{\text{probe}}} \Delta_f(x).$$

**Hyperparameters.** In our experiments, we set $n$ to be 20. Larger $n$ provides more informative embeddings but requires proportionally more forward passes, as each continuation must be scored by both the base and finetuned models. Despite this, the dimensionality of Delta Meaning remains extremely compact compared to weight- or logit-based alternatives. Importantly, because any model can evaluate the probability of a given text sequence, Delta Meaning embeddings are naturally architecture-agnostic, allowing us to cluster finetuned models drawn from multiple backbones.

# B  ADDITIONAL ANALYSIS

## B.1  BEYOND DOMAINS: CLUSTERING BY DATASET PROPERTIES.

In our setting in Section 3.1, each domain corresponds to a well-defined task with relatively uniform answer structure (e.g., multiple-choice). Here, we ablate that structure. We construct a new model pool by finetuning 3 models on each of five distinct subsets of `Tulu v2` (Ivison et al., 2023): CoT, GPT4-ALPACA, SHAREGPT, CODEALPACA, and SCIENCE. Unlike domain datasets, these are less semantically disjoint. Instead, they differ in instruction format, conversational structure, and expected output style—ranging from open-ended dialogue to multi-step chain-of-thought reasoning, code snippets, and longform factual answers. Crucially, this makes the output distribution far more diverse and less predictable. Results are presented in Table 11. With no consistent answer template, output sentence embeddings fail to reflect model specialization for LLaMA and Qwen, whereas Delta Activations continue to achieve decent clustering quality.

| Embedding Space | LLaMA | Gemma | Qwen | Avg. |
|---|---|---|---|---|
| output sentence embeddings | .06 | .23 | $-.03$ | .08 |
| Delta Activations | **.33** | **.41** | **.48** | **.41** |

Table 11: **Clustering quality (silhouette score) across Tulu v2 instruction splits.** Delta Activations remain effective despite diverse output structures and blurred instruction boundaries.

## B.2  ROBUSTNESS TO TRAINING SETTINGS.

Do differences in training settings have a greater impact on Delta Activations than the choice of finetuned domains? To evaluate this, we systematically perturbed the training process for models within our domain clusters. In our main setting, the model pool is organized into 5 distinct domain clusters, with 3 models in each cluster trained using identical settings. To test the impact of a specific training configuration—for instance, learning rate—the three models within each of the 5 domain clusters were trained using three different learning rates respectively (e.g., model 1 with $1e^{-4}$, model 2 with $4e^{-4}$, model 3 with $1e^{-5}$ within a single domain cluster). This process was independently repeated for variations in the number of training examples and the number of training epochs, where we vary number of training examples by 100, 1000, and 10000 and number of epochs by 1, 2, and 3.

Table 12 presents results which measure the clustering quality on domains when subjected to such training variations. indicates that varied training settings generally did not break domain-specific clustering. Changes to the amount of training data or the number of epochs had minimal effect on the quality of these clusters, which remained comparable to those formed under identical training settings. The different-learning-rate setting yields a lower silhouette score, as expected, since learning rate significantly impacts training dynamics and tends to increase within-cluster variation. These observations confirm that Delta Activations effectively identify finetuning domains despite common variations in training procedures. On the other hand, these results also show the strength of Delta Activations in identifying the nuanced differences within each cluster.

| Training Setting | LLaMA | Gemma | Qwen | Avg. |
|---|---|---|---|---|
| Different number of training examples | .66 | .51 | .68 | .62 |
| Different learning rates | .53 | .37 | .23 | .38 |
| Different training epochs | .62 | .59 | .51 | .57 |
| Identical training settings | .65 | .55 | .65 | .61 |

Table 12: **Delta Activations' embeddings are robust to training hyperparameters.** Models trained in varying settings still form tight domain-specific clusters, comparable to those trained identically.

### B.3 T-SNE VISUALIZATION ON MORE BACKBONES

We show visualization of the experiments conducted in Section 3.1 on LLaMA and Qwen in Figure 7 and Figure 8 respectively. Over different backbones, the visualization consistently shows the superiority of Delta Activations in forming cleanly-separated clusters.

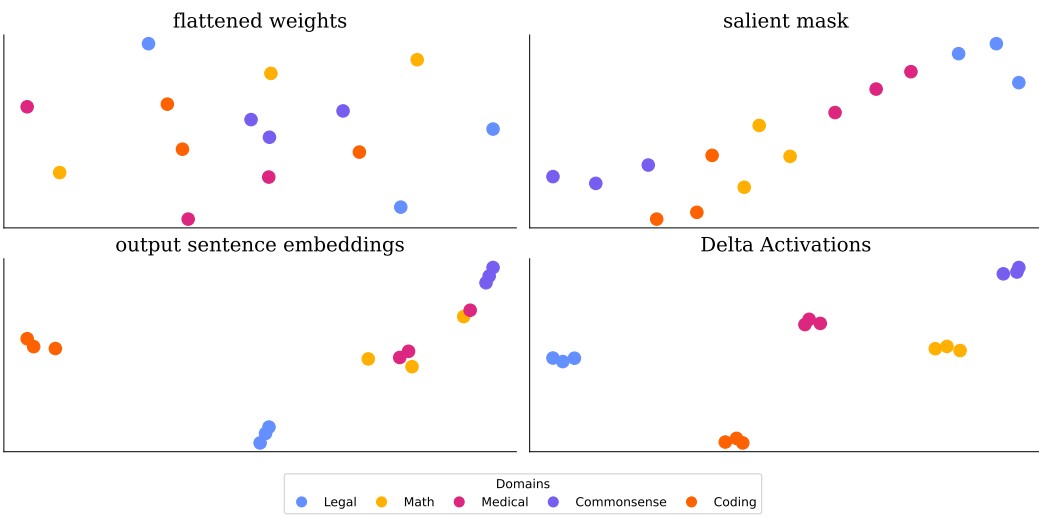

Figure 7: **t-SNE visualization of different embedding spaces (LLaMA).**

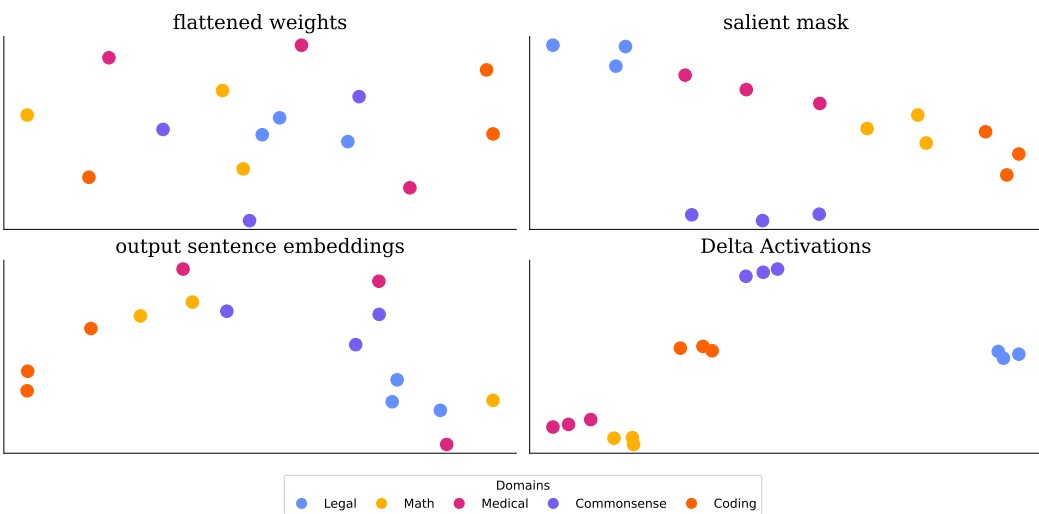

Figure 8: **t-SNE visualization of different embedding spaces (Qwen).**

### B.4 Full Results for Additive Properties experiment

To better understand the additive nature of Delta Activations, we extend the experiment from Section 3.1 to cover all ten domain pairs. In each case, we compare the Delta Activation vector from a model trained on the mixed dataset against those from models trained individually on each domain, as well as their vector sum. This comprehensive table demonstrates that the summed Delta Activations consistently better approximate the mixed-model embedding, reinforcing the additive property of Delta Activations.

| Math | Commonsense | Code | Medical | Legal | Mixed v. D1 | Mixed v. D2 | Mixed v. Sum |
|:---:|:---:|:---:|:---:|:---:|:---:|:---:|:---:|
| ✓ | | ✓ | | | .703 | .270 | **.726** |
| ✓ | ✓ | | | | .577 | .484 | **.649** |
| | ✓ | ✓ | | | .631 | .283 | **.653** |
| | | | ✓ | ✓ | .407 | .675 | **.695** |
| | ✓ | | ✓ | | .359 | .662 | **.677** |
| ✓ | | | ✓ | | .760 | .697 | **.811** |
| | | ✓ | ✓ | | .462 | .581 | **.693** |
| | ✓ | | | ✓ | .649 | .659 | **.763** |
| ✓ | | | | ✓ | .522 | .610 | **.669** |
| | | ✓ | | ✓ | .445 | .507 | **.587** |

Table 13: **Additive property (full).** The sum of Delta Activations on models finetuned separately on two datasets aligns well with Delta Activations on the model finetuned on two datasets mixed together.

### B.5 Full finetuning

We conduct experiments in Section 3.1 on LLaMA-3.1-8B with the model pool trained using full finetuning instead of LoRA. This experiment yields a silhouette score of **0.63**, which confirms that Delta Activations provide clear clustering regardless of finetuning methods. Visualization is shown in Figure 9.

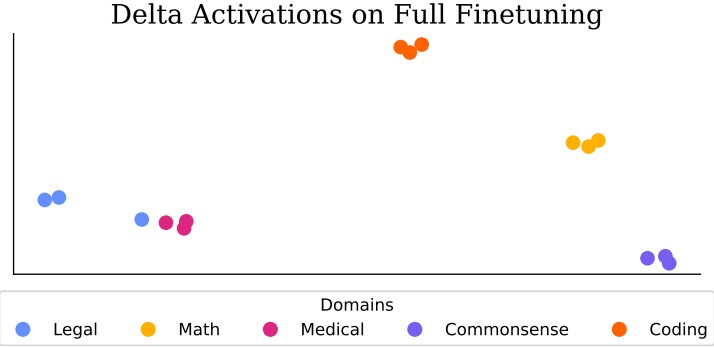

Figure 9: **t-SNE visualization of full finetuning.**

**Intra-domain clustering.** We conduct experiments to test whether Delta Activations provide sufficient signal to differentiate models trained on different sub-expertises within medical and coding domains.

We partition each domain into multiple sub-expertise splits. For medical, we create finetuned 8 models: disease databases (FreedomIntelligence, 2024) (2 splits), disease-symptoms (Mohamed-Ahmed161, 2024) (2 splits), MedQA-USMLE (Jin et al., 2020) (2 splits), and MedMCQA (Pal et al., 2022) (2 splits). For coding, we create 6 models covering C++ (2 splits), Python (2 splits), and Java (2 splits) from CodeChef (CodeChef, 2009), Codeforces (Quan et al., 2025), and Evol-Instruct (Luo et al., 2024c) datasets. We compute Delta Activations using both generic instruction prompts and domain-specialized prompts.

Table 14 shows that generic prompts achieve a silhouette score of 0.657 for medical sub-expertise, outperforming specialized medical prompts (0.533). For coding, both prompts perform comparably (0.668 vs 0.674). The generic probe maintains sufficient resolution to form correct clusters of sub-expertises within domains.

| Domain | Generic prompt | Specialized prompt |
|--------|:--------------:|:------------------:|
| Medical | **.657** | .533 |
| Coding | .668 | **.674** |

Table 14: **Intra-domain clustering.** Silhouette scores for sub-expertise clustering within domains.

## C  USE OF LARGE LANGUAGE MODELS (LLMS)

We use LLMs to polish writing, and assist in writing code for our method and preparing demo.

## D  PROMPT TEMPLATES

Table 15, Table 16, and Table 17 list all prompt templates used in probe dataset experiments in Section 3.2. For other experiments, we use the first five prompts in Table 15.

| ID | Prompt Template |
|----|-----------------|
| 1 | Below is an instruction that describes a task. Write a response that appropriately completes the request. |
| 2 | The task described below requires a response that completes the request accurately. |
| 3 | Below is a description of a task. Provide a response that aligns with the requirements. |
| 4 | The following instruction outlines a task. Generate a response that meets the specified request. |
| 5 | You are given an instruction and input. Write a response that completes the task as requested. |
| 6 | You are provided with a task instruction and input. Write a response that fulfills the described requirements. |
| 7 | Here is an instruction and its associated input. Complete the task with an appropriate response. |
| 8 | Below is a task along with its context. Write a response that matches the requirements. |
| 9 | The following is a description of a task and its input. Generate a response that fulfills the request. |
| 10 | An outlined task is provided along with its input. Write a response that satisfies the given instruction. |
| 11 | Given the following instruction, generate a suitable response that fulfills the request. |
| 12 | The task described below requires a response that completes the request accurately. |
| 13 | Below is a description of a task. Provide a response that aligns with the requirements. |
| 14 | The following instruction outlines a task. Generate a response that meets the specified request. |
| 15 | You are given an instruction and input. Write a response that completes the task as requested. |
| 16 | Here is an instruction and its associated input. Create a response that properly addresses the request. |
| 17 | Below is a task description. Provide an appropriate response that matches the input. |
| 18 | An instruction and input are provided. Write a response that accurately completes the task. |
| 19 | The following is an instruction that describes a task. Write a response that correctly satisfies the request. |
| 20 | Below is an outlined task. Respond with a completion that fits the instruction and input given. |

Table 15: List of paraphrased prompt templates used in our experiments.

| ID | Prompt Template |
|----|-----------------|
| 1 | Instruction: Please provide a response. Input: Input. |
| 2 | Please perform the following task. |
| 3 | Complete the instruction. |
| 4 | Provide the appropriate response. |
| 5 | Here is the text. Response: |

Table 16: One-sentence paraphrased prompt templates.

| ID | Prompt Template |
|----|-----------------|
| 1 | Response: |
| 2 | Answer: |
| 3 | Explanation: |
| 4 | Solution: |
| 5 | Discussion: |

Table 17: One-word paraphrased prompt templates.

## E  BROADER IMPACT

The proposed Delta Activations method facilitate efficient reuse of fine-tuned models by providing an embedding to encode the finetuned model's behaviors and capability. This reduces redundant training, cutting energy costs and promoting sustainable AI practices. Furthermore, it encourages broader public sharing of fine-tuned models by offering clear documentation of their capabilities, accelerating research and collaboration.

However, expanding public model hubs also introduces risks, as low-quality or adversarial models could contaminate the pool. This highlights the need for careful curation to maintain reliability and safety in open model ecosystems.

