# OpenReview forum: "Delta Activations: A Representation for Finetuned Large Language Models"
_ICLR.cc/2026/Conference — ICLR 2026 Conference Withdrawn Submission_

### Official Review · Reviewer_9JVg · 2025-10-31

**Soundness:** 2
**Presentation:** 2
**Contribution:** 2
**Rating:** 4
**Confidence:** 3

**Summary:**

This paper introduces "Delta Activations", a simple and efficient method to create a compact vector "fingerprint" for any fine-tuned LLM. The method works by feeding a small set of generic, task-agnostic prompts into both the fine-tuned model and its original base model, and then calculating the _difference_ (the delta) between their internal activations at the final layer. The authors demonstrate that these DA fingerprints are highly effective, allowing models to be automatically and accurately clustered by their specialized domain. Furthermore, the paper shows that this embedding space has some potential capabilities, like additive property and model selection.

**Strengths:**

1. Delta Activations method is simple and efficient. It only requires model inference.
2. The paper discusses various potential methods and shows that Delta Activations works best.
3. The paper points out potential further research directions.

**Weaknesses:**

1. The motivation is not clear enough. Why representation of models in the same pool should be close? I feel it's more like a hypothesis assumption. The model with very close embedding to a specific task may have bad generalization.
2. The paper's experimental results are mainly based on the silhouette score, which is just a "proxy" metric measuring how well the embeddings clustered. However, the main objective should be applications like downstream task performance, while this paper rarely shows such results.
3. **Model selection and similarity measurement** (line 421) paragraph is the only place that shows downstream task results. However, this experimental setup is a bit vague. Why only identify the _single_ most-related model and sample the remaining 19 models randomly? Why not sample the top 20 most-related models, which is more aligned to the paper's hypothesis and should even show further improvement?
4.  **Additive property** experiments show Mixed and Sum has high similarity. But what's the benefit here is unclear. This setting has a big gap between model merging. And a maximum 0.73 cosine similarity in table 4 is not very strong.
5. The experimental setup is too ideal. The data domains are separated clearly, and they are more independent. But real training data is usually mixed and complex.

**Questions:**

1. Could you explain more about **Additive property** paragraph in line 290? I don't get what's the benefit there.
2. In line 210, it should be **each** pool contains 15 models or it includes 15 models in total for three pools?

---

### Official Review · Reviewer_6MYB · 2025-10-31

**Soundness:** 2
**Presentation:** 2
**Contribution:** 3
**Rating:** 4
**Confidence:** 4

**Summary:**

Fine-tuned large language models (LLMs) are abundant but hard to reuse due to poor metadata and disorganization. This paper proposes **Delta Activations**—a lightweight way to turn fine-tuned models into vector embeddings by comparing internal activation differences between fine-tuned and base models on generic prompts.

It outperforms baselines in domain-based clustering (average silhouette score 0.614 across 3 base models) and has key strengths: robustness to training changes, additive properties for mixed datasets, and extension into the **Delta-X framework** (supporting logits/semantic representations). It also enables few-shot task embedding for model retrieval and better model selection for merging (2.0% BBH accuracy gain), aiding efficient reuse of public fine-tuned LLMs.

**Strengths:**

1.	An interesting problem. It points out the difficulty of reusing fine-tuned LLMs caused by messy metadata and unorganized repositories.
2.	Lightweight and efficient: Delta Activations only needs one forward pass to compute, avoiding complex calculations like matrix factorization.
3.	Strong clustering ability: It outperforms baselines (e.g., flattened weights, output embeddings) in grouping models by fine-tuned domains, with an average silhouette score of 0.614 across three base models.

**Weaknesses:**

1.	Insufficient evidence for research motivation: The paper claims fine-tuned LLMs are underused due to poor metadata, but lacks real-world data (e.g., stats on unused models or user surveys) to prove this problem.
2.	Vague practical applications for reuse: It mentions aiding model reuse, but gives few details on how end-users (e.g., developers) would actually apply it, like no step-by-step example of retrieving a model for a real task.
3.	Relies on internal model access: It needs hidden activations, which are unavailable for closed-source LLMs—limiting its real-world use where many LLMs are proprietary.

**Questions:**

1.	Could you provide more real-world evidence (e.g., statistics on unused fine-tuned LLMs, surveys of developers’ reuse struggles) to support the scale and urgency of the "poor metadata causing underused models" problem?
2.	Can you give a concrete, step-by-step example of how end-users (e.g., a developer building a medical app) would apply Delta Activations to retrieve and reuse a domain-specialized fine-tuned model?

---

### Official Review · Reviewer_KPrH · 2025-11-01

**Soundness:** 2
**Presentation:** 4
**Contribution:** 2
**Rating:** 2
**Confidence:** 4

**Summary:**

This paper proposes a new method called "Delta Activations," which aims to create efficient vector representations (embeddings) for a vast collection of fine-tuned LLM that often lack metadata.

The method works by feeding a fixed set of generic probe datasets into both a fixed "base model" and a "fine-tuned model." It then calculates the difference (the "delta") between their last-layer hidden states. By aggregating these different vectors (e.g., by averaging), a unique vector embedding is generated for the fine-tuned model.

**Strengths:**

* The method shows superior performance when clustering models that are initialized from the same base model.

* The paper is well-written and flows smoothly.

**Weaknesses:**

The paper's main contribution, "Delta Activations," has a fundamental limitation: it heavily relies on a shared, architecturally identical base model. This is because the method's core operation is the calculation of differences between high-dimensional activation vectors (e.g., 4096-D). However, in today's LLM ecosystem, models are often based on different architectures or are closed-source, making their internal architectures inaccessible. Therefore, the truly critical and pressing challenge is cross-architecture model representation and clustering.

The paper relegates this key challenge to an extension called "Delta Meaning." This approach represents a massive compromise: it degenerates from a high-dimensional (4096-D) internal activation space to an extremely low-dimensional (20-D) external probabilistic space. As shown in Table 3, the representational power of Delta Meaning is far weaker than that of Delta Activations (a score of just 0.20 vs. 0.61), confirming that a significant amount of critical internal information is lost during the transition to this probabilistic space.
In essence, Delta Activations solves a "simple problem" that relies on overly strong assumptions and is limited in real-world scenarios. Meanwhile, the solution it provides for the "real problem" (cross-architecture clustering) is excessively compromised on performance. Consequently, it is not a convincing solution for organizing a heterogeneous model ecosystem.

Furthermore, if the model zoo is very large, even a single inference pass per model will introduce significant computational overhead. The method also relies on a fixed, small, "generic" Probe Dataset.

**Questions:**

What is the biggest and most critical application scenario for this method?

---

### Note · Authors · 2026-01-05

I have read and agree with the venue's withdrawal policy on behalf of myself and my co-authors.